# Evaluating Dependencies in Fact Editing for Language Models: Specificity and Implication Awareness

**Zichao Li, Ines Arous, Siva Reddy, Jackie C.K. Cheung**
Mila, McGill University
{zichao.li,ines.arous,siva.reddy}@mila.quebec
jackie.cheung@mcgill.ca

## Abstract

The potential of using a large language model (LLM) as a knowledge base (KB) has sparked significant interest. To manage the knowledge acquired by LLMs, we need to ensure that the editing of learned facts respects internal logical constraints, which are known as *dependency of knowledge*. Existing work on editing LLMs has partially addressed the issue of dependency, when the editing of a fact should apply to its lexical variations without disrupting irrelevant ones. However, they neglect the dependency between a fact and its logical implications. We propose an evaluation protocol with an accompanying question-answering dataset, DepEdit, that provides a comprehensive assessment of the editing process considering the above notions of dependency. Our protocol involves setting up a controlled environment in which we edit facts and monitor their impact on LLMs, along with their implications based on If-Then rules. Extensive experiments on DepEdit show that existing knowledge editing methods are sensitive to the surface form of knowledge, and that they have limited performance in inferring the implications of edited facts.[1]

## 1 Introduction

Recent advancements in Large Language Models (LLMs) have sparked interest in using them as knowledge bases (KBs) due to their high performance in recalling factual knowledge (Petroni et al., 2019a; Cohen et al., 2023b). This growing interest is mainly motivated by the advantages that LLMs offer over traditional knowledge bases. First, LLMs require minimal human supervision as opposed to KBs that must be manually populated. Second, LLMs process queries expressed in natural language, unlike KBs, which require users to input queries with a specific syntax.

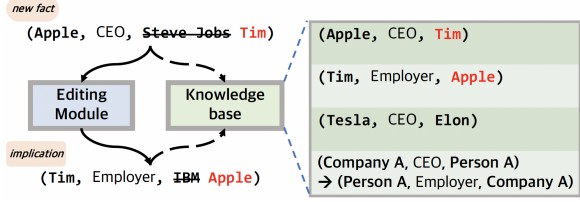

Figure 1: The editing of symbolic KBs is *specific*: unrelated fact *(Tesla, CEO, Elon)* is kept unchanged; and *implication-aware*: an new implication *(Tim, Employer, Apple)* is added to the KB accordingly.

While the potential of LLMs as KBs is recognized, their reliability remains contentious. The data used to train LLMs may contain inaccuracies or outdated information, compromising the trustworthiness of using LLMs as KBs. Consequently, it is imperative to enable practitioners to edit LLMs, ensuring that the knowledge[2] in LLMs can be rectified and reflects current real-world knowledge.

When editing knowledge in LLMs, it is crucial to ensure that the modifications respect internal logical constraints commonly referred to as *dependency of knowledge* in the database and knowledge base communities (Fagin et al., 1983; Fan, 2008; Pawlak; Ullman, 1988). Dependency of knowledge refers to the logical constraints between different pieces of information that have to be satisfied in order to ensure the knowledge base's integrity. It mainly includes specificity and implication awareness constraints. Specificity refers to editing a fact and its lexical variations, while other irrelevant ones are left unaltered. Implication awareness ensures that the editing process effectively derives implications using *If-Then* rules from an edited fact. For instance, in Figure 1, editing a fact *(Apple, CEO, Steve Jobs)* by replacing 'Steve Jobs' with

---

[1]Code and data: https://github.com/McGill-NLP/LogicalKnowEdit

[2]*Knowledge of LLMs* in this paper refers to the model's beliefs that are acquired from the training data without justification in the real world. In the realm of epistemology, this type of knowledge is referred to as weak knowledge (Goldman and Olsson, 2009).

'Tim Cook' should keep the unrelated fact *(Tesla, CEO, Elon)* unchanged, while the implication *(Tim Cook, Employer, IBM)* should be updated to *(Tim Cook, Employer, Apple)*.

While fundamental questions related to dependency have been settled in the knowledge base community (Fagin et al., 1983; Fan, 2008; Pawlak; Ullman, 1988; Arenas et al., 1999; Calì et al., 2003; Bohannon et al., 2006), they are still in their early stage in LLMs-as-KBs research. Since knowledge acquired by LLMs is encoded within a set of interconnected neurons, editing one fact is likely to impact others. Moreover, LLMs do not explicitly represent the dependencies between different pieces of knowledge, which makes the consequences of any edit unknown in advance. Recent work (Meng et al., 2022b; Mitchell et al., 2021; De Cao et al., 2021) tackled the challenge of editing knowledge in LLMs but only addressed the specificity constraint. However, they neglect the implication awareness of editing, which is a core part of the dependency constraints in editing knowledge.

In this paper, we address both the specificity and implication awareness of knowledge editing in LLMs. Concretely, we propose a new evaluation protocol called *establish-and-update* and an accompanying English question-answering dataset (`DepEdit`) for a comprehensive evaluation of dependency in editing LLMs. The evaluation is done by creating an extensive collection of knowledge editing simulations. Each simulation focuses on a subset of the dataset, where we are able to control the entities, relations and applicable *If-Then* rules described within this subset. Consequently, we compare the expected and actual edits applied by a knowledge editing model and evaluate its specificity and implication awareness.

Using our dataset `DepEdit` and evaluation protocol, we compare state-of-the-art knowledge editing methods in LLMs. The results illustrate the challenges associated with implication-aware editing. We find that existing methods tend to be influenced by superficial patterns such as the surface form of questions. Furthermore, our analysis of models' gradient updates shows that they struggle to identify the appropriate gradient direction for updating the parameters of the LLM in relation to implications. Finally, we discuss future work and show that there is significant potential in drawing inspiration from methods developed for dependency in KBs to effectively edit knowledge

in LLMs.

## 2 Related Work

**LLMs as Knowledge Bases**  Recent studies have highlighted the potential of large language models (LLMs) as knowledge bases. For example, Petroni et al. (2019b) and Davison et al. (2019) have shown that one can recall factual knowledge from LLMs through prompt-based methods without depending on external knowledge sources. However, subsequent research (Cao et al., 2021; Wang et al., 2021a) has raised concerns about the reliability of LLMs as knowledge bases, particularly regarding their sensitivity to prompts.

This work provides a different testbed to assess the reliability of LLMs as knowledge bases, focusing on the influence of editing on knowledge dependency.

**Knowledge Editing in LLMs**  Various methods have recently been proposed to edit the knowledge acquired in LLMs. One line of work (De Cao et al., 2021; Mitchell et al., 2021) develops external learned editors that modify the tuning gradients for editing, while keeping the base model unchanged. Another line of work (Meng et al., 2022a; Dai et al., 2022; Meng et al., 2022b) attributes knowledge to particular neurons or modules in the network and manually edits their activation to edit a target fact. While these methods achieve high performance in editing knowledge in LLMs, they only evaluate under the specificity constraint, i.e. editing a fact without affecting the rest of the knowledge. Our work adopts a broader perspective on knowledge editing and focuses on its capability to incorporate dependencies that encapsulates both specificity and implication awareness. In a similar vein, Hase et al. (2023), Cohen et al. (2023a) and a concurrent work (Onoe et al., 2023) also go beyond the scope of specificity and evaluate the editing of entailed facts. However, they mainly focus on single-hop reasoning and do not have *If-Then* rules to track the editing of implications.

**Dependencies in Knowledge Bases**  Dependency theory finds its roots in the domain of data dependency within relational databases (Fagin et al., 1983; Rauszer, 1984). It was later adapted to knowledge bases, as the semantic aspects of dependencies between facts and rules within a KB (Wu et al., 1994; Buszkowski and Orlowska, 1998; Pawlak). Several methods (Slota et al., 2011; Slota and Leite,

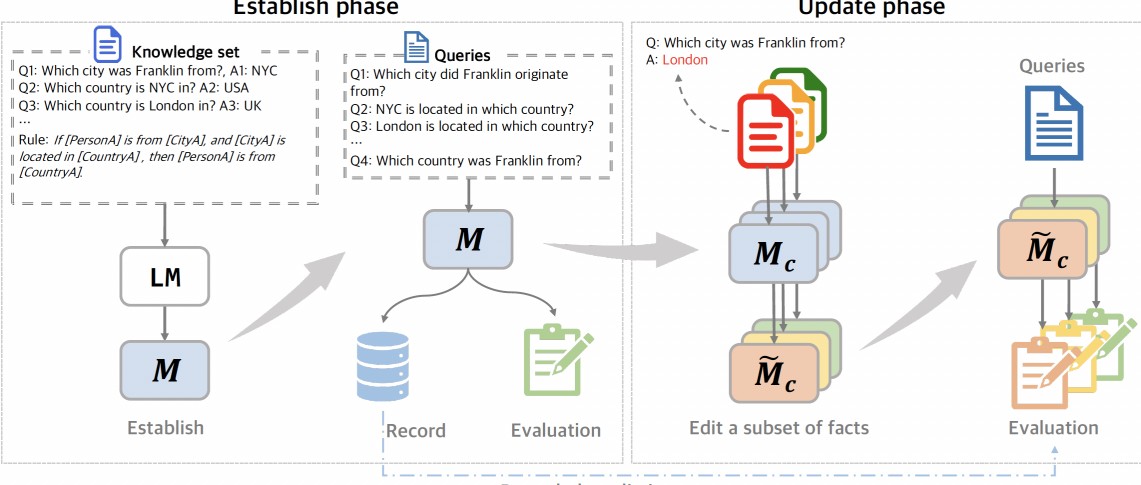

Figure 2: The complete pipeline of our *establish-and-update* evaluation protocol. During the establish phase, we prompt a model **M** to extract its knowledge about the facts and implications within a knowledge set. During the update phase, some facts are edited, and we track the change of the facts and accordingly implications in the updated model $\tilde{\mathbf{M}}_c$.

2012; Zhang and Foo, 1995) were developed for knowledge editing in KBs based on dependencies. These methods were based on key principles (Dalal, 1988), including three fundamental ones: "Irrelevance of Syntax," which ensures the meaning of edited knowledge remains independent of the syntax of the original and the updated knowledge; "Persistence of Prior Knowledge," which aims at preserving existing information during editing; and "Primacy of New Information," which prioritizes new information over existing one in the KB.

Inspired by this body of work and drawing parallels between knowledge editing in KBs and LLMs, our research aims to evaluate knowledge editing methods in LLMs on the principles mentioned above. Specifically, we map the principles of "Irrelevance of Syntax" and "Persistence of Prior Knowledge" to a specificity constraint, which evaluates an editing method's ability to edit facts and their lexical variations without disrupting irrelevant ones. We map the "Primacy of New Information" principle to an implication awareness constraint, where we evaluate a method's ability to infer logical implications. Note that existing work on knowledge editing in LLMs mainly evaluated based on the specificity constraint (De Cao et al., 2021; Mitchell et al., 2021; Dai et al., 2022), while omitting the implication awareness constraint. To the best of our knowledge, we are the first to establish such analogy and evaluate LLMs using these principles.

# 3 The Establish-and-Update Protocol for Evaluating Knowledge Editing

## 3.1 Notations

Throughout this paper, we denote the set of facts with $\mathcal{F}$ and the set of *If-Then* rules with $\mathcal{R}$. Each fact $f \in \mathcal{F}$ is defined as a triplet $(e_{f,1}, Rel_f, e_{f,2})$, where $e_{f,1}$ and $e_{f,2}$ are entities and $Rel_f$ is their relation. Each fact $f$ is mapped to a question-answer pair $(q_f, a_f) = \mathcal{T}(f)$ using a mapping table in our dataset.

**Example 1.** *The fact $f$ expressed in its triplet form $(e_{f,1}, Rel_f, e_{f,2})$ as (Franklin, City, NYC) is transformed to a question-answer pair $(q_f, a_f)$: (Which city was Franklin from?, NYC).*

Each rule $r \in \mathcal{R}$ is an *If-Then* rule composed of a logical AND between two premises $P_{r,1}$ and $P_{r,2}$, leading to an implication $I_r$. In other terms, a rule $r$ is expressed as follows:

```
If (P_{r,1} AND P_{r,2}) then I_r.
```

Both premise and implication $P_r$ and $I_r$ are expressed in a template form '$?x \, Rel \, ?y$', where $?x$ and $?y$ denote entity types, and $Rel$ their relation. They can also be instantiated with specific facts. For example, '$[PersonA]$ is from $[CityA]$' can be instantiated to '$[Franklin]$ is from $[NYC]$'.

We denote a *knowledge set* with $\mathcal{K}$, which contains subsets of facts $\mathcal{F}_k$ and rules $\mathcal{R}_k$, such that $\mathcal{K} = \mathcal{F}_k \cup \mathcal{R}_k$. The facts and rules in $\mathcal{K}$ are expressed in natural language form. The facts $\mathcal{F}_k$

in a knowledge set $\mathcal{K}$ include two types of facts: specific and unrelated facts, respectively denoted as $\mathcal{S}_k$ and $\mathcal{U}_k$. The specific facts $\mathcal{S}_k$ share the entity types and relations of the premise propositions in the rules $\mathcal{R}_k$. Unrelated facts are sampled from the facts $\mathcal{F}$ such that $\mathcal{U}_k \in \mathcal{F} \setminus \mathcal{S}_k$. From $\mathcal{S}_k$ and rules $\mathcal{R}_k$, we derive implications $\mathcal{I}_k$.

**Example 2.** *A knowledge set $\mathcal{K}$ contains If-Then rules and QA pairs mapped from specific facts, implications and unrelated facts:*

- ***Specific facts*** *$\mathcal{T}(\mathcal{S}_k)$: (Which city was Franklin from?, NYC); (Which country is NYC in?, USA); (What is the country of London?, UK)*
- ***Rules*** *$\mathcal{R}_k$: If [Person A] is from [City A], and [City A] is located in [Country A], then [Person A] is from [Country A].*
- ***Implications*** *$\mathcal{T}(\mathcal{I}_k)$: (Which country was Franklin from?, USA)*
- ***Unrelated facts*** *$\mathcal{T}(\mathcal{U}_k)$: (Who was Demi Moore's child?, Rumer Willis);*

*Note that the entities (e.g., Franklin) in $\mathcal{S}_k$ are instantiations of the entity type in $\mathcal{R}_k$ (e.g., Person A).*

We denote the changes expected after editing knowledge in a LLM with a tilde symbol (e.g., $\tilde{\mathcal{F}}$ to denote edited facts).

## 3.2 Overview

We propose to evaluate the specificity and implication awareness of knowledge editing methods in LLMs by simulating the editing process of a LLM within a controlled environment. In each simulation, we use a knowledge set $\mathcal{K}$ with facts $\mathcal{F}_k$ and rules $\mathcal{R}_k$. The simulation comprises two phases: an establish and an update phase. In the *establish* phase, we provide a knowledge editing model with the knowledge set $\mathcal{K}$ and prompt it to extract its *established knowledge*, which refers to the facts and rules it has learned. In the *update* phase, we edit a subset of the facts. Then, we compare the edits made by the model with the expected edits. An overview of the two phases is presented in Figure 2.

Following De Cao et al. (2021) and Mitchell et al. (2021), we choose question answering (QA) as the base task. Nevertheless, our evaluation protocol is not specific to QA and could be applied to other tasks. Next, we describe each phase followed by our evaluation metrics.

## 3.3 Establish Phase

The goal of the establish phase is to extract the facts and rules a model learns given a knowledge set $\mathcal{K}$. The model is free to use arbitrary methods (including doing nothing) to establish the given knowledge. We then prompt the established model $\mathbf{M}$ to evaluate its performance in terms of recalling the facts and implications from $\mathcal{K}$. We record the model's predictions and corresponding questions as its *established knowledge* and denote it with $\mathcal{Q}_k = \{(q, \mathbf{M}(q))\}$. The established knowledge $\mathcal{Q}_k$ includes what the model recalls as specific facts $\mathcal{Q}_S$, unrelated facts $\mathcal{Q}_U$, and implications $\mathcal{Q}_I$.

## 3.4 Update Phase

The update phase evaluates the ability of a LLM to update its knowledge given new information. We proceed by first creating copies, denoted $\mathbf{M}_c$, of the previously established model. Then, for each copy $\mathbf{M}_c$, we edit some specific facts and their implications, while other specific facts are kept unchanged. We denote the expected edits on specific facts as $\tilde{S}$ and on implications $\tilde{I}$. The model is then updated with $\tilde{S}$ to $\tilde{\mathbf{M}}_c$. For each updated model, we use the established knowledge $\mathcal{Q}_k$ from the establish phase to assess whether the non-updated facts and implications remain unchanged.

**Example 3.** *Let's take the case where a model learns from the knowledge set in Example 2. Given a new fact in the update phase: (Which city did Franklin originate from?, London), we evaluate a knowledge editing model ability to edit lexical variations such as (Franklin was from which city?, London) and derive implications such as (Which country was Franklin from?, UK)*

## 3.5 Evaluation Metrics

We use the *exact-match* score (EMS) as a metric to measure the rate at which a knowledge editing method prompted with a question $q$ returns the expected answer $a$. The EMS is given by:

$$\text{EMS}(\mathbf{M}, \mathcal{D}) = \frac{1}{|\mathcal{D}|} \sum_{(q,a) \in \mathcal{D}} \mathbb{1}[\mathbf{M}(q) = a], \quad (1)$$

where $\mathcal{D}$ denotes the subset of question-answer pairs $(q, a)$, and $\mathbb{1}[\cdot]$ is an indicator function returning 1 if the model $\mathbf{M}$ prompted with $q$ returns the answer $a$ and 0 otherwise.

**Establish phase** For the establish phase, we evaluate the established model $\mathbf{M}$ using two metrics:

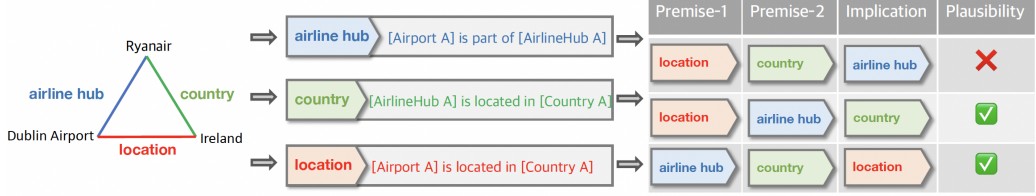

Figure 3: The Step 1 (left), 3 (middle), 5 (right) of the collection pipeline for `DepEdit`. In this example, the second candidate rule annotated by workers is: `If [Airport A] is located in [Country A], and [Airport A] is part of [AirlineHub A], then [AirlineHub A] is located in [Country A]`.

the establish success rate on specific facts (**Est.S** [3]), and their implications (**Est.I**). We compute:

$$\text{Est.S} = \text{EMS}(\mathbf{M}, \mathcal{T}(\mathcal{S}))$$
$$\text{Est.I} = \text{EMS}(\mathbf{M}, \mathcal{T}(\mathcal{I})),$$

**Update phase** At the end of the update phase, we evaluate a model's $\mathbf{M}_c$ ability to edit knowledge based on dependency constraints, which are specificity and implication awareness. To evaluate a model's specificity, we use two metrics: **Upd.S** and **Cons.NS**. The first metric, **Upd.S**, aims at evaluating a model's ability to edit a fact and its lexical variations. While, **Cons.NS** and **Cons.U**, evaluate the disruption of non-updated specific facts and unrelated facts, respectively. They are expressed as follows:

$$\text{Upd.S}_c = \text{EMS}(\tilde{\mathbf{M}}_c, \mathcal{T}(\tilde{S}))$$
$$\text{Cons.NS}_c = \text{EMS}(\tilde{\mathbf{M}}_c, \mathcal{Q}_S)$$
$$\text{Cons.U}_c = \text{EMS}(\tilde{\mathbf{M}}_c, \mathcal{Q}_U),$$

We also evaluate the implication awareness of a model using two metrics: success rate of updated implications (**Upd.I**) and consistency of non-updated implications (**Cons.NI**). We define them as follows:

$$\text{Upd.I}_c = \text{EMS}(\tilde{\mathbf{M}}_c, \mathcal{T}(\tilde{I}))$$
$$\text{Cons.NI}_c = \text{EMS}(\tilde{\mathbf{M}}_c, \hat{\mathcal{Q}}_I).$$

where $\hat{\mathcal{Q}}_I$ denotes the established implications without the updated ones. Finally, we aggregate and average the values of each updated model: $\text{Metric} = \sum_c \text{Metric}_c$.

# 4 New Knowledge Editing Dataset: `DepEdit`

Existing datasets (Mitchell et al., 2021; Meng et al., 2022b) for knowledge editing lack *If-Then* rules,

which are crucial to evaluate the implication awareness of a model. In order to facilitate models' evaluation based on both specificity and implication awareness, we introduce `DepEdit`, a new dataset for evaluating edits in language models using the zsRE dataset (Levy et al., 2017). In our dataset, we collect facts expressed both as a question-answer pair and in a triplet form i.e., $(e_{f,1}, Rel_f, e_{f,2})$ consisting of an entity pair $(e_{f,1}, e_{f,2})$ and their relation $Rel_f$ drawn from WikiData (Vrandečić, 2012). We also enumerate a large set of potentially applicable rules, then hire crowd-sourcing workers to annotate their plausibility. The most plausible rules are included in `DepEdit`. In the following, we explain each step of the dataset curation process and then provide key statistics about our dataset.

## 4.1 Dataset Curation

**Step 1** The zsRE dataset contains facts expressed in their triplet form which contains a relation between two entities. We iterate over each pair of relations in the zsRE dataset and search for a new relation via WikiData database[4]. We aim to find a new relation in WikiData that forms a clique with the pair of relations in the zsRE dataset based on shared entities.

**Step 2** We collect, using crowdsourcing, questions about the newly collected relations from Wiki-Data. (See Appendix A for more details).

**Step 3** For QA pairs sharing the same relation, we hire workers to provide a template form of the QA, such that the relation is described in natural language and the entities are replaced by their types. For instance, (*Q: In which country is Calgary located?*, *A: Canada*) is transformed to `[City A] is located in [Country A]`.

**Step 4** Using the collected templates, we generate rules by grouping relations in the same clique mined at Step 1. Then, we insert their correspond-

---

[3]We are focusing on the evaluation of a specific knowledge set. Therefore, we have omitted the knowledge set index for clarity.

[4]We use the official WikiData query engine following their regulations.

| | Questions $q_n$ | Rephrased Questions $q'_n$ | Answers ($a_n$) |
|---|---|---|---|
| Specific facts | Which city was **Franklin D. Roosevelt** from? | Which city did **Franklin D. Roosevelt** originate from? | **New York City** |
| | Which country is **New York City** in? | **New York City** is located in which country? | USA |
| Unrelated facts | Who was Demi Moore's child? | | Rumer Willis |
| *If-Then* rule | If [**Person A**] is from [**City A**], and [**City A**] is located in [**Country A**], then [**Person A**] is from [**Country A**]. | | |
| Implications | Which country was **Franklin D. Roosevelt** from? | | USA |

Table 1: A knowledge set from `DepEdit`. We show two specific facts and one unrelated facts. Entities(e.g. USA) and their corresponding entity type(e.g. [Country A]) in rules are in bold and share the same color. Refer to Table 8 for more examples.

ing templates into an *If-Then* rule, where two relations act as premises and one relation serves as an implication. As a result, we generate three rules for each clique, resulting in a total of approximately 7k generated rules.

We do not require rules to be logical entailments, as we assume that a controllable knowledge base should be able to adhere to rules provided by a user. However, we would like the rules in our dataset to correspond to plausible rules that could be given in a deployment setting. We thus hire annotators to filter out implausible rules, such as those that contradict commonsense knowledge (e.g., "*[Person A] is from [Country A], and [City A] is the capital of [Country A], then [Person A] was born in [City A]*").

**Step 5** For each candidate rule, two annotators rate its plausibility by determining given the premises, the consequence: *(a) Must be true*; *(b) Likely to be true*; *(c) Unlikely to be true*; *(d) Must be false*; or *(e) The given conditions are not useful for the derivation of the consequence*. We adopt the rules on which both of workers agree that their plausibility are (a) or (b).

The key steps 1, 3 and 5 are illustrated in Figure 3.

## 4.2 Crowdsourcing Tasks

The crowdsourcing tasks involved in steps 2, 3 and 5 were published on Amazon Mechanical Turk[5], where workers generated questions, template forms of QA pairs and filtered implausible rules. We hired workers from English-speaking countries with $> 95\%$ approval rate that had passed qualification test. Each task took between 20-40 sec to complete on average. Workers who completed the task received a reward corresponding to an hourly wage around 12–20 USD. Details on the setup and quality control of each annotation task are included in Appendix A.

[5]https://www.mturk.com/

## 4.3 Key Statics on `DepEdit`

`DepEdit` contains 159,264 question-answer pairs, each associated with its corresponding triplet form. Additionally, the dataset includes a total of 530 *If-Then* rules. Using the QA pairs and the *If-Then* rules, we create knowledge sets as in Example 2. Consequently, we obtain 1106 knowledge sets. We show an example of these sets in Table 1.

## 5 Experiments

### 5.1 Task Setup

We split `DepEdit` into 1106 groups of simulations, each corresponding to one knowledge set. Each knowledge set contains 29 facts (24 specific facts and 5 unrelated facts), 1 If-Then rule, and 12 implications. During the update phase, we create three new versions of facts, where we randomly sample 20 facts from a knowledge set and edit their answers. Note that it is possible to include multiple rules per simulation. However, we find that the current setting with one rule is already challenging for existing methods (see Section 5.3.2).

| Settings | Questions for editing | Questions for evaluation |
|---|---|---|
| **CQ_DT** | Which city was Franklin from? J. K. Rowling was from which city? | Which city was Franklin from? J. K. Rowling was from which city? |
| **CQ_UT** | Which city was Franklin from? Which city was J. K. Rowling from? | Which city was Franklin from? Which city was J. K. Rowling from? |
| **ICQ_DT** | Which city was Franklin from? J. K. Rowling was from which city? | Which city did Franklin originate from? Which city was J. K. Rowling from? |

Table 2: Examples of specific facts and prompted questions in different settings.

We evaluate knowledge editing methods based on dependency constraints, including specificity and implication awareness. We use three settings to evaluate the specificity of editing methods that we name: **CQ_DT**, **CQ_UT** and **ICQ_DT**. We use the term **CQ**(**c**onsistent **q**uestioning) to refer

to a setting where we use the same randomly sampled question templates for specific facts during editing (i.e., establish and update) as during evaluation, while **ICQ** (**i**nconsistent **q**uestioning) is used to describe a setting where they are different. We use the term **UT** (**u**niform **t**emplate) to describe a setting where we use the same question templates for each relation in specific facts within a knowledge set, and the templates are selected randomly. On the other hand, **DT** (**d**iverse **t**emplates) is used when different question templates are used. Table 2 illustrates each setting. These settings are designed to test for the principle of "Irrelevance of Syntax". In particular, **ICQ** evaluates if the editing is consistent across different linguistic presentations of a specific fact. Meanwhile, **UT** tests whether the editing method can distinguish different pieces of knowledge that share similar surface forms.

To evaluate a method's implication awareness, we evaluate a method's ability to answer questions about implications.

## 5.2 Comparison Methods

We compare with two pre-trained language models, BART (Lewis et al., 2020) as used by De Cao et al. (2021); Mitchell et al. (2021) and GPT-XL (Radford et al., 2019) as used by Meng et al. (2022b), which serve as baselines for knowledge editing methods. To mitigate the domain disparity, we finetune GPT-XL on a closed-book QA dataset (Wang et al., 2021b). We also compare with the following knowledge editing methods:

- **FINETUNE**: continuously fine-tune the base model whenever new knowledge is given.
- **MEND** (Mitchell et al., 2021): learns an additional hypernetwork that updates the parameters of a language model, while satisfying two constraints: the edits apply to lexical variations, and answers to irrelevant questions remain unaltered.
- **MEMIT** (Meng et al., 2022b): applies causal mediation tracing on a GPT model to identify and alter the most significant layers for a fact.
- **RANDOM**: is a trivial baseline that randomly sample from a set of answer candidates seen during establish and update phase.

Additional implementation details can be found in Appendix B.

## 5.3 Results

We use our establish-and-update protocol to evaluate existing methods' ability to edit knowledge,

based on specificity and implication awareness.

### 5.3.1 Specificity of editing

The results for establish and update phases are presented in Table 3. First, we observe that MEMIT has a higher success rate than MEND in both the establish and update phases. Second, we observe a significant gap (more than $10\%$) in **Est.S** and **Upd.S** between MEMIT and FINETUNE. Third, we observe that MEND and MEMIT have better performance than FINETUNE in terms of consistency on unrelated facts(**Cons.U**). This finding is consistent with the literature (Mitchell et al., 2021; De Cao et al., 2021; Meng et al., 2022b). However, MEMIT performs worse than FINETUNE in terms of consistency of non-updated specific facts (**Cons.NS**).

When comparing the different settings, we find that the performance of all methods, except BART and GPT-XL, decreases in the **ICQ_DT** compared with **CQ_DT**. This result indicates the difficulty that existing methods face in handling lexical variations during the editing process. When comparing **CQ_DT** and **CQ_UT**, we observe that the performance of MEND decreases (e.g. $\approx 4.5\%$ in terms of **Upd.S**), while the performance of other methods stays almost constant. This result indicates that using questions with similar surface form deteriorates the performance of MEND.

| Setting | Methods | Est.S(%)[↑] | Upd.S(%)[↑] | Cons.NS(%)[↑] | Cons.U(%)[↑] |
|---|---|---|---|---|---|
| - | RANDOM | 4.70 | 5.06 | 4.22 | 5.21 |
| Base model: BART | | | | | |
| CQ_DT | BART | 7.49 | 0.37 | 100. | 100. |
| | FINETUNE | 90.85 | 89.88 | 36.64 | 15.00 |
| | MEND | 34.19 | 47.79 | 65.43 | 54.71 |
| CQ_UT | BART | 7.43 | 0.40 | 100. | 100. |
| | FINETUNE | 88.62 | 87.69 | 32.57 | 15.17 |
| | MEND | 31.67 | 44.32 | 47.51 | 53.60 |
| ICQ_DT | BART | 7.59 | 0.46 | 100. | 100. |
| | FINETUNE | 62.41 | 61.20 | 20.16 | 15.00 |
| | MEND | 27.08 | 41.14 | 53.65 | 54.71 |
| Base model: GPT-XL | | | | | |
| CQ_DT | GPT-XL | 0.06 | 0.00 | 100.0 | 100.0 |
| | FINETUNE | 98.66 | 99.28 | 77.36 | 40.38 |
| | MEND | 33.51 | 49.50 | 98.29 | 64.15 |
| | MEMIT | 34.14 | 85.16 | 58.06 | 94.15 |
| CQ_UT | GPT-XL | 0.06 | 0.00 | 100.0 | 100.0 |
| | FINETUNE | 97.33 | 97.98 | 66.13 | 40.00 |
| | MEND | 28.74 | 43.26 | 95.98 | 63.21 |
| | MEMIT | 34.02 | 84.31 | 57.38 | 93.53 |
| ICQ_DT | GPT-XL | 0.07 | 0.00 | 100.0 | 100.0 |
| | FINETUNE | 85.91 | 95.72 | 59.69 | 40.38 |
| | MEND | 20.70 | 32.14 | 87.95 | 64.15 |
| | MEMIT | 14.36 | 56.38 | 56.84 | 94.15 |

Table 3: Performance of different knowledge editing methods in terms of *specificity* during establish (**Est.S**) and update phase(**Upd.S**, **Cons.NS**, **Cons.U** ).

### 5.3.2 Implication awareness of editing

We employ a variant of FINETUNE called FINETUNE$_{\text{RULE}}$, where we fine-tune a LM model by generating an implication conditioned on its two premises. For MEND and MEMIT, we edit the premises and test whether they update the implications according to the edited premises.

Table 4 presents the results. We observe that MEND and MEMIT achieve low performance in terms of **Est.I** and **Upd.I**. Meanwhile, they achieve high **Cons.NI** at the same level of their **Cons.U** in Table 3, which suggests that these editing methods regard the implications as unrelated knowledge. Therefore, both MEND and MEMIT do not establish logical dependencies when editing knowledge. The establish and update success rate is the same for both FINETUNE and FINETUNE$_{\text{RULE}}$, which means that incorporating the *If-Then* rules with FINETUNE$_{\text{RULE}}$ does not improve the performance of the FINETUNE model. This finding suggests that LMs do not internalize *If-Then* rules expressed in natural language.

| Methods | Establish phase | Update phase | |
| --- | --- | --- | --- |
| | **Est.I**(%)$^\uparrow$ | **Upd.I**(%)$^\uparrow$ | **Cons.NI**(%)$^\uparrow$ |
| RANDOM | 8.41 | 8.65 | 8.62 |
| Base model: BART | | | |
| BART | 11.59 | 7.15 | 100.0 |
| FINETUNE | 17.80 | 6.53 | 16.69 |
| FINETUNE$_{\text{RULE}}$ | 17.84 | 6.57 | 18.40 |
| MEND | 14.92 | 10.99 | 47.08 |
| Base model: GPT-XL | | | |
| GPT-XL | 0.10 | 0.00 | 100.0 |
| FINETUNE | 19.99 | 9.70 | 38.44 |
| FINETUNE$_{\text{RULE}}$ | 19.99 | 9.70 | 38.44 |
| MEMIT | 0.54 | 0.68 | 94.15 |
| MEND | 12.91 | 11.73 | 62.04 |

Table 4: Performance of different rule learning methods and knowledge editing methods in terms of *implication awareness* during establish and update phase.

## 6 Analysis

Next, we conduct analysis to investigate the barriers to implication-aware knowledge editing. Formally, we break down this task into two sub-goals: (1) finding the gradient direction $\nabla_\theta P_\theta(a_I|q_I)$ for the LLM's parameters $\theta$ to reflect the editing of implication conditioned on the update incurred by new facts $\nabla_\theta P_\theta(a_f|q_f)$. (2) finding the optimal gradient direction to simultaneously update the facts and implications. We start by discussing the second

| Gradient pairs | Cosine similarity |
| --- | --- |
| Premises & Implication | 0.52 |
| Fact pairs with similar surface form | 0.51 |
| Original & Rephrased facts | 0.77 |
| Random fact pairs | 0.35 |

Table 5: Gradient's similarity when updating different pairs of facts.

sub-goal before moving to the first one, since it is easier to directly estimate the difficulty of the second one.

### 6.1 Gradient Similarity between Updating Premise Facts and Implications

We assume that the similarity between the gradients of premise facts and implications, when a LM learns QA pairs related to them, serves as an indicator of the feasibility of implication-aware editing. Intuitively, the higher the similarity between these gradients, the easier it is to perform implication aware editing. Therefore, we measure the similarity between gradients for updating premise facts $\nabla_\theta \log P_\theta(a_f|q_f)$ and implications $\nabla_\theta \log P_\theta(a_I|q_I)$. For the sake of comparison, we measure the gradient similarity between updating facts with similar surface forms, pairs of facts with lexical variations, and randomly sampled ones. As seen in Table 5, the gradient similarity of premise-implications pair is on par with pairs of facts with similar surface form. This suggests that updating a fact and its logical implication is as challenging as updating facts with similar surface form.

### 6.2 Gradient Direction to Update Implications

The above analysis suggests that the main challenge lies in the first sub-goal: determining the direction to update the parameters following the applicable rules. To further validate this hypothesis, we train MEND to capture the logical dependencies among premise facts and implications. We proceed by modifying its learning objective so that it applies the edit for implications in a similar manner as it applies to the lexical variation of facts. The details of the modification can be found in Appendix C.

We refer this variant as MEND$_{\text{IMP}}$. To further verify that the difficulty mainly comes from identifying the direction to update the gradients

| Methods | **Est.I**(%) | **Ed.I**(%) |
| --- | --- | --- |
| MEND | 12.61 | 10.77 |
| MEND$_{\text{IMP}}$ | 14.01 | 9.64 |
| BART$_{\text{DERIVE}}$ | 84.59 | 84.35 |

Table 6: Performance of MEND variants update and edit implications.

conditioned on the
update of premises,
we setup another approach: BART$_{\text{DERIVE}}$, where the premise facts are given in the prompt. Concretely, we prepend both the premises and randomly sampled unrelated facts before the implications questions $q_I$ for the model. As seen in Table 6, MEND$_{\text{IMP}}$ achieves low performance despite explicitly encoding the logical dependencies in its training. In contrast, BART$_{\text{DERIVE}}$ achieves much higher performance, revealing that inferring an implication given its premises is not challenging when trained to predict implications given premises.

## 6.3 Dependency Resolution with ForwChain and BackChain

To accurately measure the disparity between current methods and flawless dependency resolution, we introduce two additional toplines: FORWCHAIN and BACKCHAIN inspired by forward-chaining and backward-chaining algorithms used in traditional KB systems (Al-Ajlan, 2015). Both methods assume that we can infer questions about implications from questions about their premises. BACKCHAIN breaks down the question into inquiries about the premises based on applicable rules, while FORWCHAIN updates the KB by querying the language models for other premises whenever a new fact is provided. We apply these methods with FINETUNE and MEMIT for GPT-XL.

The results in Table 7 show that the external inference mechanism improves the accuracy of editing implications. The accuracy gain with MEMIT is smaller than FINETUNE. However, FORWCHAIN and BACKCHAIN introduce new risks that can potentially decrease the consistency of non-updated implications. With BACKCHAIN, the model needs to give consistent answers to both of the premises questions, which can compound any existing inconsistencies. FORWCHAIN increases the number of premise facts to be updated, making the editing easier to break irrelevant knowledge.

| Methods | External Inference | Establish phase Est.I(%) | Update phase Upd.I(%) | Cons.I(%) |
|---|---|---|---|---|
| FINETUNE | FORWCHAIN | - | 86.28(85.6) | 39.15(0.71) |
| FINETUNE | BACKCHAIN | 98.21(78.22) | 76.79(67.09) | 38.85(0.41) |
| MEMIT | FORWCHAIN | - | 22.06(21.38) | 81.11(−13.04) |
| MEMIT | BACKCHAIN | 14.19(13.65) | 30.59(29.91) | 62.97(−31.17) |

Table 7: Performance (and performance gain) in terms of *implication awareness* with our protocol.

## 7 Conclusion

In this paper, we propose a novel evaluation protocol designed to assess knowledge dependency, including both the specificity and implication awareness in knowledge editing methods applied to LLMs. To implement our evaluation protocol, we collect a new question-answering dataset `DepEdit` with If-then rules in natural language. Our experiments reveal limitations in current knowledge editing techniques, both in terms of specificity and their ability to understand implications. In general, state-of-the-art knowledge editing methods exhibit a lower success rate compared to vanilla fine-tuning and are highly influenced by the precise wording of facts. Additionally, current methods demonstrate limited performance in deriving the logical consequences of facts using natural language rules. Future work could draw inspiration from existing methods in KB editing and leverage our dataset and evaluation protocol to develop implication aware knowledge editing methods.

## 8 Limitations

Our evaluation protocol is inspired by the principles of knowledge editing in knowledge bases (Dalal, 1988). These principles are more comprehensive than the dependency constraints discussed in the paper. For instance, the principle of "primacy of new information" applies to all potential logical operations, such as quantification and probabilistic entailment. Currently, `DepEdit` does not support the evaluation of knowledge editing in LLMs with such complex logical operations. Furthermore, each knowledge set in the current setting of our experiments contains only one rule. We carefully allocate the edit targets so that there are no conflicts among the given facts and derived implications, which may occur in real-world applications. That said, we find that the existing knowledge editing methods fail even in this simple setting. Future work can extend our `DepEdit` dataset by aggregating multiple rules in one knowledge set or assigning probability values to rules and facts.

## 9 Ethics Statement

A large portion of question-answer pairs in `DepEdit` are transformed from WikiData triplets, which could be inaccurate. Furthermore, the If-Then rules we collected are labeled by a small group of annotators. These points mean that there

is no guarantee that the facts and the rules are factual, and they may reflect stereotypes or biases. Therefore, `DepEdit` can only be used for testing knowledge editing and other reasoning abilities of computational models, not for fact-checking, decision making, or other similar purposes.

## 10 Acknowledgement

This work was supported by Samsung Electronics, the NSERC Discovery Grant program, and the Canada CIFAR AI Chair program. We acknowledge the material support of Nvidia and Digital Research Alliance of Canada in the form of computational resources.

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

## A Dataset collection details

### A.1 Mining Cliques of Relations and Question Collection

Starting from all the pairs of triplets included in the zsRE (Levy et al., 2017) dataset, we extracted those pairs with overlapped entities. For instance, $(e_1, r_1, e_2)$ and $(e_1, r_2, e_3)$ has an overlapped entities $e_1$. Then, we queried the WikiData engine to search potential relations $r_3$ between $e_2$ and $e_3$. After that, we continued this process by searching new relations conditioned on $r_1, r_3$ and $r_2, r_3$, respectively. Through this process, we obtained around 2500 relation triangles.

For the new relations not presented in zsRE (Levy et al., 2017), we hired crowd-sourcing workers to transform the triplet $(e_1, r_1, e_2)$ to a question-answer pair such that we obtain new questions about $e_1$ and $r_1$, and a corresponding answer $e_2$. For instance, (RedOctane, copyright representative, Activision) is transformed to *Q: What is the copyright representative of RedOctane?, A: Activision.* We collected roughly 900 QA pairs for about 150 relations. We manually filtered out the ones of low quality. Common errors include incorrect interpretation of relations and grammar errors. Finally, more than 700 QA pairs are included.

The reward of this task was set to be $0.15 per Human Intelligence Task (HIT), where each one took less than 20 seconds to complete on average.

### A.2 Question-Answer pairs summarization and Potential Rules Construction

We asked the workers to summarize sets of QA pairs grounded on the same relations to abstractive and descriptive templates that summarize the relations between two classes of entities. The task was decomposed into two sub-tasks: (1) rewrite these two groups of entities to their abstract form respectively, and (2) describe the relation between these two abstract forms. Take the following groups of QA pairs as an example: *Q: Which city is the capital of [Canada/USA/UK/India/Germany]; A: [Ottawa/Washington, D.C./New Delhi/Berlin]*, the template can be `[CityA] is the capital of [CountryA]`. The workers were asked to summarize according to the following instructions:

- Format of abstract forms: A word or a phrase that describes the category of the entities. The first letter of each word is upper case and there

is an uppercase letter ($A$ or $B$) in the end as an index to distinguish different entities. That means, if the categories of two entity groups are the same, the index letter should be $A$ and $B$ respectively. Otherwise, both of them should be $A$.

- The template should be a description of the relation that the questions ask about, and it should contain the rewritten abstract form without mentioning a specific entity.

- The template should be descriptive. That is, it should end with a period instead of a question mark.

The reward of this task was set to be $0.15 per HIT and each HIT took less than 40 seconds to complete.

We used JavaScript to ensure that the submissions strictly follow the above constraints. After collecting the templates, we pluged the templates of the relations on the same triangle into an If-Then formula with two premises and one implication. For each relations triangle, there were three potential If-Then rules. Finally, we had around 7k potential rules after this step.

### A.3 Evaluation of plausibility of rules

Before starting the formal task, workers were asked to take part in a qualification test, annotating a small subset that was labeled by us. More than 100 workers took part in the qualification test. Seven top-performing workers of them were selected. For each HIT, the annotators were given a potential rule, and some entities example of the abstract forms, such as `[CapitalA]` and `[CountryA]`, and then asked to determine the logical plausibility of deriving the implication conditioned on the premises. In particular, they were asked to follow these instructions:

- Determining the plausibility of the reasoning step described by the If-Then rule, instead of the plausibility of the conditions or of the consequence themselves.

- The entity examples are only given to help you think about the rule in more concrete terms. What we care about is the general rule, not just whether it applies to those specific examples

Each rule was labeled by two workers. For collecting more reliable labels, we set up two rounds

of annotation. At the first round, about 2000 rules with at least one positive label (*must be true* or *likely to be true*) were selected for the second round. At the second round, we have more strict selection criteria: the rules should have positive labels ((a) or (b)) with perfect agreement. Finally, we have included about 530 rules in DepEdit.

## A.4 Other annotation details

All annotation tasks clarified that the data would be used to build a QA system. For the final plausibility evaluation of rules, we adopted additional qualification tests. Throughout the collection of question collection and template summarization, we blocked workers who had made low-quality submissions.

We strictly follow the research ethics protocol of our institution. The first author received human research ethics training and passed the exam. We will release DepEdit for academic usage.

## A.5 More examples in DepEdit

More examples of knowledge sets from DepEdit are presented in Table 8.

## B Experiment Details

We used the pre-trained BART-based (Lewis et al., 2020) model from De Cao et al. (2021) as the base model. It contains 110M parameters. And we followed the same hyper-parameters setting and optimizer to train MEND as Mitchell et al. (2021): Adam optimizer (Kingma and Ba, 2015) with $1r - 6$ learning rate. During preliminary experiments, we found that MEND is not robust to multi-step editing, which has also been reported in Hase et al. (2023). Therefore, in the update phase, we apply and evaluate MEND on base models fine-tuned with FINETUNE in the establish phase.

As for FINETUNE, we followed De Cao et al. (2021); Mitchell et al. (2021), using the RMSProp optimizer (Tieleman et al., 2012) with $1e - 5$ learning rate. BART was fine-tuned for 50 steps until the loss is smaller than $1e - 7$.

As for MEMIT, we use the same set of hyper-parameters from (Meng et al., 2022b) and the GPT-XL model as the base LLM (Radford et al., 2019).

The main experiments were conducted on Intel Gold 6148 Skylake CPUs and NVidia V100SXM2 GPUs. The training time of a MEND model was estimated at 24 GPU hours. Evaluation of each editing method was finished in less than 8 GPU hours.

## C Details of Experiment Setup in Section 6.2

Given specific facts $(q_f, a_f)$ and their lexical variations $(q'_f, a_f)$, unrelated facts $(q_u, a_u)$, the vanilla MEND is trained to refine the gradient $\nabla_\theta \log P_\theta(a_f|q_f)$ of a LLM's parameters $\theta$ to $\nabla^*_\theta \log P_\theta(a_f|q_f)$, and the resulting updated parameters are denoted as $\theta^*$.

$$\begin{aligned} \mathcal{L}_{\text{MEND}} = & - \log P_{\theta^*}(a_f|q'_f) \\ & + \text{KL}(P_\theta(\cdot|q_u)||P_{\theta^*}(\cdot|q_u)) \end{aligned} \quad (2)$$

Given the implications $(q_I, a_I)$ derived from specific facts and applicable rules, The learning objective of MEND$_{\text{IMP}}$ is then:

$$\mathcal{L}_{\text{MEND}_{\text{IMP}}} = \mathcal{L}_{\text{MEND}} - \log P_{\theta^*}(a_I|q_I) \quad (3)$$

We also divide DepEdit into train/validation/test sets ensuring that each set shares the same rule but contains different entity pairs. Additionally, we reduced the number of facts to six and the number of implications to three, where four of the facts and hence two implications were updated during the update phase. This approach allowed us to focus the evaluation on implication awareness, thereby mitigating the potential impact of the lack of specificity when editing large-scale facts.

| | Questions $q_n$ | Rephrased Questions $q'_n$ | Answers $(a_n)$ |
|---|---|---|---|
| | | Example 1 | |
| Specific facts | In which fictional work is **Sherlock Holmes** a character? | What piece of fiction does **Sherlock Holmes** appear in? | **The Adventure of the Three Students** |
| | The **Sherlock Holmes** was made by whom? | Which was the creator of **Sherlock Holmes**? | **Arthur Conan Doyle** |
| Unrelated facts | Who is the copyright holder of IBM PC DOS? | | IBM |
| *If-Then* rule | If [**Character A**] is from [**Novel A**], and [**Character A**] was created by [**Person A**], then [**Novel A**] was written by [**Person A**]. | | |
| Implications | Who wrote **The Adventure of the Three Students**? | | **Arthur Conan Doyle** |
| | | Example 2 | |
| Specific facts | In which country is **Calgary** located? | Which country is **Calgary** in? | **Canada** |
| | In which continent is **Calgary** located? | What continent is **Calgary** located in? | **North America** |
| Unrelated facts | What was Benjamin Franklin's occupation? | | Diplomat |
| *If-Then* rule | If [**Location A**] is located in [**Country A**], and [**Location A**] is in [**Continent A**], then [**Country A**] is located in [**Continent A**]. | | |
| Implications | Which continent is **Canada** located in? | | **North America** |

Table 8: More knowledge sets from `DepEdit`.