# OpenReview forum: "Evaluating Dependencies in Fact Editing for Language Models: Specificity and Implication Awareness"
_EMNLP/2023/Conference — EMNLP 2023 Findings_

### Official Review · Reviewer_uPob · 2023-08-03

**Soundness:** 3

**Excitement:**

2: Mediocre: This paper makes marginal contributions (vs non-contemporaneous work), so I would rather not see it in the conference.

**Paper Topic And Main Contributions:**

In this paper, the authors propose an evaluation protocol with an accompanying QA dataset, that provides a comprehensive assessment of the editing process considering the notions of dependency.

**Questions For The Authors:**

- the authors stated that each set of knowledge in the current setting of their experiments contained only one rule => How could it be extended to more rules?
- Also, they find that the existing knowledge editing methods fail even in this simple setting => What will happen when the settings will be more complex?

**Reasons To Accept:**

- good and actual references from important conferences and journals
- the formalization of the problem is well done, and every step is completed with examples
- the examples help the reader to understand better what is the problem and how the authors deal with it
- the authors propose an evaluation protocol that can be used in different contexts

**Reasons To Reject:**

- their experiments show that existing knowledge editing methods are sensitive to the surface form of knowledge and that they have limited performance in inferring the implications of edited facts
- the proposed methodology is very good and detailed, but the results obtained are not up to expectations
- one of the most important parts of the proposed solution is made by the workers, but we do not have details about the quality of the work done by them and about the trust in what they did

**Reproducibility:**

3: Could reproduce the results with some difficulty. The settings of parameters are underspecified or subjectively determined; the training/evaluation data are not widely available.

**Reviewer Confidence:**

3: Pretty sure, but there's a chance I missed something. Although I have a good feel for this area in general, I did not carefully check the paper's details, e.g., the math, experimental design, or novelty.

**Typos Grammar Style And Presentation Improvements:**

Line 478: "unrelated facts(Cons.U)." => "unrelated facts (Cons.U)."

Lines 494-495 Table 3: "and update phase(Upd.S, Cons.S, Cons.U )." => "and update phase (Upd.S, Cons.S, Cons.U)."

Line 868: "on r1," => 1 need to be underscript

Lines 895-897: "Q: Which city is the capital of [Canada/USA/UK/India/Germany]; A: [Ottawa/Washington, D.C./New Delhi/Berlin]," => "Q: Which city is the capital of [Canada/USA/UK/India/Germany]; A: [Ottawa/Washington, D.C./London/New Delhi/Berlin],"

Line 950: "likely to be true)were" => "likely to be true) were"

Line 967: "ation(in both formal" => "ation (in both formal"

---

> ### Author Rebuttal · Authors · 2023-08-27
>
> Thank you for recognizing our comprehensive literature review, clarity in our writing and presentation, and the usefulness of the proposed evaluation protocol.
>
> ### Reasons-To-Reject-1+2 (experiments shows the weaknesses of existing knowledge editing methods, results obtained are not up to expectations)
>
> Thank you for recognizing that our evaluation protocol reveals the limitations of existing editing methods. The unexpected results are interesting findings that inspire the community to address these limitations. While we conducted analysis in Section 6  to identify the bottleneck of maintaining dependencies, proposing a concrete solution, though important, is out of the scope of this paper.
>
> ### Reasons-To-Reject-3 (details about workers’ performance)
>
> We adopted several methods for quality control, each of which is detailed in the paper.
>
> * Annotators selection criterions: Workers were hired based on their English proficiency and approval rate(>95%) (detailed in Lines 393-399). We also conducted additional qualification tests, where candidates were asked to annotate a subset of rules that had been pre-labeled by the author, and the top-performing workers were selected. More details can be found in Lines 926–930.
> * Detailed Instruction: We provided comprehensive walk-through instructions for each annotation task (see Sections A.2 & A.3).
> * Annotation Post-Processing: Low-quality annotations are either filtered automatically (Lines 901–918) or manually (Lines 880–883), and only the rule candidates that have perfect agreement are used (Lines 946–954).
>
> ### Q1&Q2 (extend the current setting with more rules)
>
> One can extend the setting with more than one rule by simply aggregating different knowledge sets. Maintaining dependencies during knowledge editing becomes more challenging in this context, since the editing method must take into account a larger number of rules, and therefore, more potential logical implications. We have already discussed this scenario in the Limitations section (see Lines 638-648).
>
> Nevertheless, we would like to emphasize that our evaluation protocol is designed to be general, and it does not impose any restrictions on the number of rules (see Lines 220-230).

---

### Official Review · Reviewer_UeeZ · 2023-08-04

**Soundness:** 3

**Excitement:**

3: Ambivalent: It has merits (e.g., it reports state-of-the-art results, the idea is nice), but there are key weaknesses (e.g., it describes incremental work), and it can significantly benefit from another round of revision. However, I won't object to accepting it if my co-reviewers champion it.

**Paper Topic And Main Contributions:**

This paper proposes an establish-and-update evaluation protocol for assessing the specificity and implication awareness of knowledge editing methods in LLMs.

**Questions For The Authors:**

Q1. The authors point out that for each candidate rule, two annotators rate its plausibility, and they adopt the rules for which the annotators agree on their plausibility. However, it is not specified how the rules that both annotators disagree on are handled, and there is no information on how to evaluate the consistency among annotators. Additionally, the identity of the annotators is not mentioned.

Q2. How was the annotation conducted? Was it done using a self-developed platform or through another form?

Q3. It is recommended to provide an illustrative annotation process diagram to facilitate readers' understanding.


**Reasons To Accept:**

This paper focuses on exploring the dependency between a fact and its logical implications, which holds significant importance in the context of Fact Editing for LLMs.

**Reasons To Reject:**

1. To some extent, the establish-and-update method is relatively simple.

2. Some details of the annotation process are missing.

Please refer to the "Questions For The Authors".

**Reproducibility:**

3: Could reproduce the results with some difficulty. The settings of parameters are underspecified or subjectively determined; the training/evaluation data are not widely available.

**Reviewer Confidence:**

4: Quite sure. I tried to check the important points carefully. It's unlikely, though conceivable, that I missed something that should affect my ratings.

---

> ### Author Rebuttal · Authors · 2023-08-27
>
> Thanks for recognizing the importance of the dependency notions adopted by our work for Editing LLMs.
>
> ### Reasons-To-Reject-1 (method is relatively simple)
>
> We see the simplicity of our evaluation protocol as a strength, as it allows for easy and flexible adoption in future work.
>
> ### Reasons-To-Reject-2 (missing annotation details)
>
> The key steps of data construction are illustrated in Section 4.1. We provide a detailed response to your questions about specific details as follows.
>
> * **Q1 (how the rules that both annotators disagree on are handled)**
>
> As detailed in Lines 946-954, we used only the rules on which workers agree concerning their plausibility, while discarding any rules that garnered disagreement. Consequently, the consistency (Cohen’s κ) between annotators regarding the adopted rules is 1. We will move this information into the main body in the final version. This agreement-based filtering method is widely used in other commonsense plausibility datasets (Zhang+, Emami+, Pradhan+).
>
> * **Q2**
>   * **Identity of reviewers and annotation platform** Thank you for your reminder! We hired annotators from Amazon Mturk platform, and we will add this information in the paper. The annotators were from English speaking countries with more than 95% approval rate (mentioned in Lines 393-399).
>
>   * **How was the annotation conducted** For the annotation of rule plausibility, we set up a qualification test prior to the launch of the formal task. Over 100 workers who met the above criteria (from certain countries and with high approval rates) on Mturk participated in this test, and seven top-performing workers were chosen. For further details, please refer to Section A.3. In the formal task, each rule candidate was rated in terms of its logical plausibility by two of these selected workers and then filtered based on the workers’ agreement (explained in Lines 393-399 and detailed in Lines 946-954).
>
> * **Q3 (provide an illustrative annotation process diagram)**
>
> Thank you for your suggestion! Figure 3 in the appendix illustrates the key steps during the annotation process, we will move it to Section 4 in the final version of the paper.

---

### Official Review · Reviewer_A8T6 · 2023-08-06

**Typos Grammar Style And Presentation Improvements:** Nothing worth mentioning here.
**Soundness:** 3

**Excitement:**

3: Ambivalent: It has merits (e.g., it reports state-of-the-art results, the idea is nice), but there are key weaknesses (e.g., it describes incremental work), and it can significantly benefit from another round of revision. However, I won't object to accepting it if my co-reviewers champion it.

**Missing References:**

None.

**Paper Topic And Main Contributions:**

This paper presents a new task for LLMs, known as knowledge editing, together with a novel benchmark for evaluation (and/or fine-tuning) build around Wikidata, called Standup.

By knowledge editing, the authors understand a kind of semi-supervised and/on online learning process. It can roughly be broken into a few steps:
a) initial fine-tuning of LLMs
b) "retrieval" of relational facts from LLMs -- querying with prompts LLMs to retrieve relational triples
c) "repairing" or validating generation errors, against a number of rules
d) re-using corrected facts to further fine-tune the LLM

The authors then carry out a number of experiments to demonstrate that knowledge editing, and in particular steps c) and d) give rise to large gains in performance for the knowledge/relational retrieval (sub)task.

**Questions For The Authors:**

See above.

**Reasons To Accept:**

The proposed task and the dataset described are novel.

**Reasons To Reject:**

While the paper has its merits, this reviewer found it hard to read. In fact, several readings were necessary to understand the basic intuitions behind the task and methodology proposed.

Above all, the authors fail to sufficiently explain or justify the why of this new task and benchmark, assuming that --somehow-- knowledge retrieval and knowledge editing are self evident and important. Nowhere in the paper can a good explanation regarding e.g. the practical importance of their approach be found.

It is kind to be expected that iterative fine-tuning of an LLM will progressively yield better and better results (this is a property of all foundational models, not only LLMs). The interesting contribution is the self-supervised or semi-supervised, where validation of the generated triples is done automatically, with the help of your corpus (vs validation by humans). But this is not clearly stated.

I also want to point out that you don't address or discuss the downside of i) fine-tuning an LLM for a single task, or even of ii) iterative fine-tuning: catastrophic forgetting. While increase in performance likely argues against catastrophic forgetting for ii), what about the degradation of the MTL abilities of large LLMs? Maybe you thought about this, as I see that you list models of moderate size (GPT-XL stands for GPT-2-XL, right, a model of ~1.3B parameters), but you don't really mention it. You method would likely not be appropriate for say GPT-3 -- although it is likely a model like GPT-3 (and larger) what would make the most sense as a knowledge retrieval engine.



**Reproducibility:**

2: Would be hard pressed to reproduce the results. The contribution depends on data that are simply not available outside the author's institution or consortium; not enough details are provided.

**Reviewer Confidence:**

2: Willing to defend my evaluation, but it is fairly likely that I missed some details, didn't understand some central points, or can't be sure about the novelty of the work.

---

> ### Author Rebuttal · Authors · 2023-08-27
>
> Thank you for recognizing the novelty of our evaluation protocol and dataset. We would like to highlight that other reviewers have also recognized the significance of dependency constraints (R2) in knowledge editing, as well as the clarity of our writing and presentation (R3).
>
>
> ### Reasons-To-Reject-1 (justification of knowledge editing and dependencies constraint in terms of practical usefulness)
>
> **Importance of knowledge editing**
>
> Knowledge editing is a fundamental operation that one needs to use a LM as a knowledge base. It has already been well attested in prior work(see De+,  Mitchell+ cited in the Related work section, Lines 134-157). We also provided a brief introduction in terms of its importance in the Introduction section (Lines 27-38).
>
> **Importance of dependency constraints in knowledge editing**
>
> Our work focuses on the dependency among pieces of knowledge during editing,  which is essential for maintaining the integrity of a knowledge base (mentioned in Line 57). As illustrated by Fig. 1, once the dependencies are established as a set of rules, users can update a knowledge base whenever a new event (e.g. a change of CEO) happens, without having to manually enumerate all of its logical implications.
>
> ### Reasons-To-Reject-2 (contribution not clearly stated)
>
> Thanks for recognizing that our evaluation protocol and dataset enable us to verify the dependencies in knowledge editing without costly human evaluation. We will emphasize this point further in the final version of the paper.
>
> ### Reasons-To-Reject-3 (catastrophic forgetting & degradation of MTL performance)
>
> We agree that editing knowledge in LLMs might indeed disrupt other factual knowledge, which is why we have adopted consistency metrics (Cons.S, Cons.I, Cons.U; detailed in Sec 3.5) to address the issue of catastrophic forgetting. Our evaluation protocol is applicable to any LLM, including GPT-3. Regarding the impact on other downstream tasks, we agree that this is an interesting direction to investigate for future work; however, it falls outside the scope of our paper.

---

### Meta-Review · Area_Chair_WVVF · 2023-09-19

**Recommendation:** 3

**Metareview:**

This paper presents a new task (derived from wikidata) for LLMs knowledge editing, which measures the dependency of knowledge -- whether the editing of learned facts respects internal logical constraints. The eval has two metrics (measuring lexical variation and the disruption of irrelevant facts), and has 2 phases:
1. The establish phase prompts a model to extract its knowledge about the facts and implications within a knowledge set.
2. The update phase edits some facts and tracks the change of the facts and accordingly implications in the updated model.

The paper adopt a semi-supervised online learning process:  a) initial fine-tuning of LLMs  b) "retrieval" of relational facts from LLMs  c) "repairing" or validating generation errors, against a number of rules  d) re-use corrected facts to further fine-tune the LLM
Experiments with  baselines (continuous finetuning, MEND, MEMIT) and 2 LLMs (BART, GPT-XL) shows that  the baselines do not establish logical dependencies when editing knowledge, and future research is needed to improve this.

Strength:
1. The proposed task and the dataset described are novel, and would likely be impactful for future work in knowledge editing
2. The experiment and analysis is comprehensive.

Weakness:
1. The reviewers also like to know more details about the quality of the ratings.
2. The reviewers have some concerns over the importance of knowledge editing in general, and the downside of iterative finetuning. I guess these are not specific to this paper but the research area in general.
3. The result is restricted to only one dataset

---

### Decision · Program_Chairs · 2023-10-07

**Decision:**

Accept-Findings

**Comment:**

This paper presents a new task (derived from wikidata) for LLMs knowledge editing, which measures the dependency of knowledge -- whether the editing of learned facts respects internal logical constraints. The eval has two metrics (measuring lexical variation and the disruption of irrelevant facts), and has 2 phases:
1. The establish phase prompts a model to extract its knowledge about the facts and implications within a knowledge set.
2. The update phase edits some facts and tracks the change of the facts and accordingly implications in the updated model.

The paper adopt a semi-supervised online learning process:  a) initial fine-tuning of LLMs  b) "retrieval" of relational facts from LLMs  c) "repairing" or validating generation errors, against a number of rules  d) re-use corrected facts to further fine-tune the LLM
Experiments with  baselines (continuous finetuning, MEND, MEMIT) and 2 LLMs (BART, GPT-XL) shows that  the baselines do not establish logical dependencies when editing knowledge, and future research is needed to improve this.

Strength:
1. The proposed task and the dataset described are novel, and would likely be impactful for future work in knowledge editing
2. The experiment and analysis is comprehensive.

Weakness:
1. The reviewers also like to know more details about the quality of the ratings.
2. The reviewers have some concerns over the importance of knowledge editing in general, and the downside of iterative finetuning. I guess these are not specific to this paper but the research area in general.
3. The result is restricted to only one dataset